# The Neural MMO Platform for
# Massively Multiagent Research

**Joseph Suarez**
MIT | jsuarez@mit.edu

**Yilun Du**
MIT

**Clare Zhu**
Stanford

**Igor Mordatch**
Google Brain

**Phillip Isola**
MIT

## Abstract

Neural MMO is a computationally accessible research platform that combines large agent populations, long time horizons, open-ended tasks, and modular game systems. Existing environments feature subsets of these properties, but Neural MMO is the first to combine them all. We present Neural MMO as free and open source software with active support, ongoing development, documentation, and additional training, logging, and visualization tools to help users adapt to this new setting. Initial baselines on the platform demonstrate that agents trained in large populations explore more and learn a progression of skills. We raise other more difficult problems such as many-team cooperation as open research questions which Neural MMO is well-suited to answer. Finally, we discuss current limitations of the platform, potential mitigations, and plans for continued development.

## 1 Introduction

The initial success of deep Q-learning [1] on Atari (ALE) [2] demonstrated the utility of simulated games in reinforcement learning (RL) and agent-based intelligence (ABI) research as a whole. We have since been inundated with a plethora of game environments and platforms including OpenAI's Gym Retro [3], Universe [4], ProcGen [5], Hide&Seek [6], and Multi-Agent Particle Environment [7; 8], DeepMind's OpenSpiel [6], Lab [9], and Control Suite [10], FAIR's NetHack learning environment [10], Unity ML-Agents [11], and others including VizDoom [12], PommerMan [13], Griddly [14], and MineRL [15]. This breadth of tasks has provided a space for academia and industry alike to develop stable agent-based learning methods. Large-scale reinforcement learning projects have even defeated professionals at Go [16], DoTA 2 [17], and StarCraft 2 [18].

Intelligence is more than the sum of its parts: most of these environments are designed to test one or a few specific abilities, such as navigation [9; 10], robustness [5; 13], and collaboration [6] – but, in stark contrast to the real world, no one environment requires the full gamut of human intelligence. Even if we could train a single agent to solve a diverse set of environments requiring different modes of reasoning, there is no guarantee it would learn to combine them. To create broadly capable *foundation policies* analogous to foundation models [19], we need multimodal, cognitively realistic tasks analogous to the large corpora and image databases used in language and vision.

Neural MMO is a platform inspired by Massively Multiplayer Online games, a genre that simulates persistent worlds with large player populations and diverse gameplay objectives. It features game systems configurable to research both on individual aspects of intelligence (e.g. navigation, robustness, collaboration) and on combinations thereof. Support spans 1 to 1024 agents and minute- to hours-long time horizons. We first introduce the platform and address the challenges that remain in adapting reinforcement learning methods to increasingly general environments. We then demonstrate Neural MMO's capacity for research by using it to formulate tasks poorly suited to other existing environment and platforms – such as exploration, skill acquisition, and learning dynamics in variably sized populations. Our contributions include Neural MMO as free and open source software (FOSS) under the MIT license, a suite of associated evaluation and visualization tools, reproducible baselines, a demonstration of various learned behaviors, and a pledge of continued support and development.

35th Conference on Neural Information Processing Systems (NeurIPS 2021), Sydney, Australia.

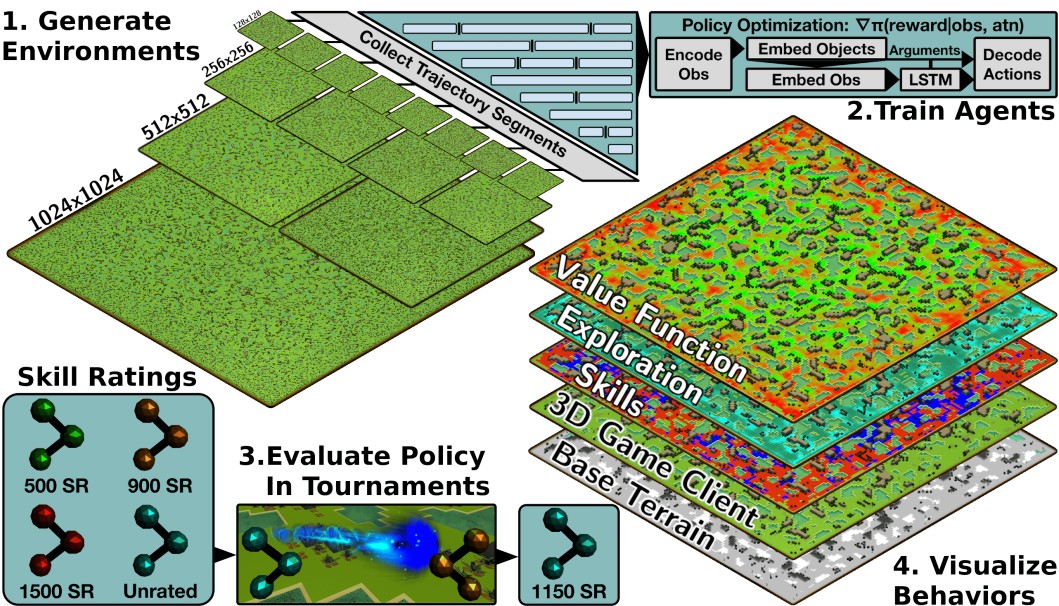

**1. Generate Environments**

128x128
256x256
512x512
1024x1024

Collect Trajectory Segments

**Policy Optimization:** $\nabla\pi(\text{reward}|\text{obs}, \text{atn})$

Encode Obs
Embed Objects
Arguments
Decode Actions
Embed Obs
LSTM

**2. Train Agents**

**Skill Ratings**

500 SR
900 SR
1500 SR
Unrated

**3. Evaluate Policy In Tournaments**

1150 SR

Value Function
Exploration
Skills
3D Game Client
Base Terrain

**4. Visualize Behaviors**

Figure 1: A simple Neural MMO research workflow suitable for new users (advanced users can define new tasks, customize map generation, and modify core game systems to suit their needs)
1. Select one of our default task configurations and run the associated procedural generation code
2. Train agents in parallel on a pool of environments. We also provide a scripting API for baselines
3. Run tournaments to score agents against several baseline policies concurrently in a shared world
4. Visualize individual and aggregate behaviors in an interactive 3D client with behavioral overlays.

## 2 The Neural MMO Platform

Neural MMO simulates populations of agents in procedurally generated virtual worlds. Users tailor environments to their specific research problems by configuring a set of game systems and procedural map generation. The platform provides a standard training interface, scripted and pretrained baselines, evaluation tools for scoring policies, a logging and visualization suite to aide in interpreting behaviors, comprehensive documentation and tutorials, and an active community Discord server for discussion and user support. Fig. 2 summarizes a standard workflow on the platform, which we detail below.

**neuralmmo.github.io** hosts the project and all associated documentation. Current resources include an installation and quickstart guide, user and developer APIs, comprehensive baselines, an archive of all videos/manuscripts/slides/presentations associated with the project, additional design documents, a list of contributors, and a history of development. Users may compile documentation locally for any previous version or commit. We pledge to maintain the community Discord server as an active support channel where users can obtain timely assistance and suggest changes. Over 400 people have joined thus far, and our typical response time is less than a day. We plan to continue development for at least the next 2-4 years.

### 2.1 Configuration

**Modular Game Systems:** The Resource, Combat, Progression, and Equipment & NPC systems are bundles of content and mechanics that define gameplay. We designed each system to engage a specific modality of intelligence, but optimal play typically requires simultaneous reasoning over multiple systems. Users write simple config files to enable and customize the game systems relevant to their application and specify map generation parameters. For example, enabling only the resource system creates environments well-suited to classic artificial life problems including foraging and exploration; in contrast, enabling the combat system creates more obvious conflicts well-suited to team-based play and ad-hoc cooperation.

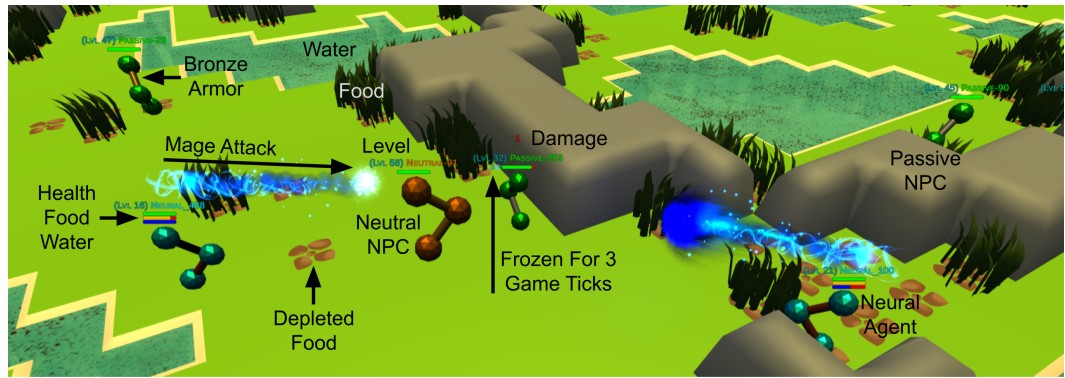

Figure 2: Perspective view of a Neural MMO environment in the 3D client

**Procedural Generation:** Recent works have demonstrated the effectiveness of procedural content generation (PCG) for domain randomization in increasing policy robustness [20]. We have reproduced this result in Neural MMO (see Experiments) and provide a PCG module to create *distributions* of training and evaluation environments rather than a single game map. The algorithm we use is a novel generalization of standard multi-octave noise that varies generation parameters at different points in space to increase visual diversity. All terrain generation parameters are configurable, and we provide full details of the algorithm in the Supplement.

**Canonical Configurations:** In order to help guide community research on the platform, we release a set of config files defining standard tasks and commit to maintaining a public list of works upon them. Each config is accompanied by a section in Experiments motivating the task, initial experimental results, and a pretrained baseline where applicable.

## 2.2 Training

**User API:** Neural MMO provides `step` and `reset` methods that conform to RLlib's popular generalization of the standard OpenAI Gym API to multiagent settings. The reward function is -1 for dying and 0 otherwise by default, but users may override the `reward` method to customize this training signal with full access to game and agent state. The `log` function saves user-specified data at the end of each agent lifetime. Users can specify either a single key and associated metric, such as "Lifetime": `agent.lifetime`, or a dictionary of variables to record, such as "Resources":{"Food": 5, "Water: 0}. Finally, an `overlay` API allows users to create and update 2D heatmaps for use with the tools below. See Fig. 3 for example usage and `neuralmmo.github.io` for the latest API.

**Observations and Actions:** Neural MMO agents observe sets of *objects* parameterized by discrete and continuous *attributes* and submit lists of *actions* parameterized by lists of discrete and object-valued *arguments*. This parameterization is flexible enough to avoid major constraints on environment development and amenable to efficient serialization (see documentation) to avoid bottlenecking simulation. Each observation includes 1) a fixed crop of *tile* objects around the given agent parameterized by *position* and *material* and 2) the other *agents* occupying those tiles parameterized by around a dozen properties including current *health*, *food*, *water*, and *position*. Agents submit *move* and *attack* actions on each timestep. The *move* action takes a single *direction* argument with fixed values of *north*, *south*, *east*, and *west*. The *attack* action takes two arguments: *style* and *target*. The *style* argument has fixed values of *melee*, *range*, and *mage*. The agents in the current observation are valid *target* argument values. Encoding/decoding layers are required to project the hierarchical observation space to a fixed length vector and the flat network hidden state to multiple actions. We also provide reusable PyTorch subnetworks for these tasks.

**Efficiency and Accessibility:** A single RTX 3080 and 32 CPU cores can train on 1 billion observations in just a few days, equivalent to over 19 *years* of real-time play using RLlib's *synchronous* PPO implementation (which leaves the GPU idle during sampling) and a fairly simple baseline model. The environment is written in pure Python for ease of use and modification even beyond the built-in configuration system. Three key design decisions enable Neural MMO to achieve this result. First, training is performed on the partially observed game state used to render the environment,

but no actual rendering is performed except for visualization. Second, the environment maintains an up-to-date serialized copy of itself in memory at all times, allowing us to compute observations using a select operator over a flat tensor. Before we implemented this optimization, the overhead of traversing Python object hierarchies to compute observations caused the entire environment to run 50-100x slower. Finally, we have designed game mechanics that enable complex play while being simple to simulate. [1] See `neuralmmo.github.io` for additional design details.

## 2.3 Evaluating Agents

Neural MMO tasks are defined by a reward function on a particular environment configuration (as per above). Users may create their own reward functions with full access to game state, including the ability to define per-agent reward functions. We also provide two default options: a simple survival reward (-1 for dying, 0 otherwise) and a more detailed achievement system. Users may select between *self-contained* and *tournament* evaluation modes, depending on their research agenda.

**Achievement system:** This reward function is based on gameplay milestones. For example, agents may receive a small reward for obtaining their first piece of armor, a medium reward for defeating three other players, and a large reward for traversing the entire map. The tasks and point values themselves are clearly domain-specific, but we believe this achievement system has several advantages compared to traditional reward shaping. First, agents cannot farm reward [3] – in contrast to traditional reward signals, each task may be achieved only once per episode. Second, this property should make the achievement system less sensitive to the exact point tuning. Finally, attaining a high achievement score somewhat guarantees complex behavior since tasks are chosen based upon difficulty of completion. We are currently running a public challenge that requires users to optimize this metric. [2].

**Self-contained** evaluation pits the user's agents against copies of themselves. This is the method we use in our experiments and the one we recommend for artificial life work and studies of emergent dynamics in large populations. It is less suitable for benchmarking reinforcement learning algorithms because agent performance against clones is not indicative of overall policy quality.

**Tournament** evaluation solves this problem by instead pitting the user's agents against different policies of known skill levels. We recommend this method for direct comparisons of architectures and algorithms. Tournaments are constructed using a few user submitted agents and equal numbers of several scripted baselines. We run several simulations for a fixed (experiment dependent) number of timesteps and sort policies according to their average collected reward. This ordering is used to estimate a single real number skill based on an open-source ranking library [3] for multiplayer games. We scale this skill rating (SR) estimate such that, on any task, our scripted combat bot scores 1500 with a difference of 100 SR indicating a 95 percent win rate. Users can run tournaments against scripted bots locally. For the next few months, we are also hosting public evaluation servers where anyone can have their agents ranked against other user-submitted agents. The neural combat agent in Table 1 scores 1150 and the scripted foraging agent scores 900. We have since improved the neural combat agent to 1600 SR through stability enhancements in the training infrastructure.

## 2.4 Logging and Visualization

Interpreting and debugging policies trained in open-ended many-agent settings can be difficult. We provide integration with WanDB that plots data recorded by the `log` function during training and evaluation. The default `log` includes statistics that we have found useful during training, but users are free to add their own metrics as well. See the documentation for sample plots. For more visual explorations of learned behaviors, Neural MMO enables users to render their 2D heatmaps produced using the `overlay` API directly within the 3D interactive client. This allows users to watch agent behaviors and gain insight into their decision making process by overlaying information from the model. For example, the top heatmap in Fig. 2 illustrates which parts of the map agents find most and least desirable using the learned value function. Other possibilities include visualizations of aggregate exploration patterns, relevance of nearby agents and tiles to decision making, and specialization to different skills. We consider some of these to interpret policies learned in our experiments.

---

[1] By adopting the standard game development techniques that enabled classic MMOs to simulate small cities of players on 90s hardware. For those interested or familiar, we leave in a few pertinent details in later footnotes.

[2] Competition page with the latest tasks: https://www.aicrowd.com/challenges/the-neural-mmo-challenge

[3] TrueSkill [21] for convenience, but one could easily substitute more permissively licensed alternatives.

| Basic Usage | Custom Reward Function | Custom Overlay (with RLlib) |
|---|---|---|

```
'''Extended OpenAI Gym Interface'''

From neural_mmo.forge.blade.io.action.static
import *
from projekt.config import CompetitionRound1

config = CompetitionRound1()
env   = CustomEnv(config)
obs   = env.reset()

actions = {}
for agentID in obs:
   players = env.realm.players
   agent   = players[agentID]

   actions[agentID] = {
      Move: {Direction: North}}

   obs, rewards, done, _ =(
      env.step(actions))
```

```
from neural_mmo.forge.trinity import Env

class CustomEnv(Env):
   '''Team Spirit shared team reward
   as proposed by OpenAI Five'''

   def reward(self, ent):
      config  = self.config
      players = self.realm.players

      dead = ent.entID not in players
      nDead = len([p for p in
         self.dead.values() if
         p.population == ent.pop])

      individual = -1 if dead else 0
      team = -nDead/config.TEAM_SIZE

      alpha = config.TEAM_SPIRIT
      return (alpha*team +
         (1.0-alpha)*individual)
```

```
from projekt.rllib_wrapper import RLlibOverlay

class Values(RLlibOverlay):
   '''Rendered in the Unity 3D Client'''
   def update(self, obs):
      players = self.realm.realm.players
      for idx, playerID in enumerate(obs):
         if playerID not in players:
            continue

         r, c = players[playerID].base.pos

         self.values[r, c] = float(
            self.model.value_function()[idx])

   def register(self, obs):
      colorized = overlay.twoTone(
         self.values[:, :])
      self.realm.register(colorized)
```

Figure 3: Neural MMO's API enables users to program in a familiar OpenAI Gym interface, define per-task reward functions, and visualize learned policies with in-game overlays.

# 3   Game Systems

The base game representation is a grid map[4] comprising grass, forest, stone, water, and lava tiles. Forest and water tiles contain resources; stone and water are impassible, and lava kills agents upon contact. At the same time, this tile-based representation is important for computational efficiency and ease of programming – see the supplementary material for a more detailed discussion.

**Resources:** *This system is designed for basic navigation and multiobjective reasoning.* Agents spawn with food, water, and health. At every timestep, agents lose food and water. If agents run out of food or water, they begin losing health. If agents are well fed and well hydrated, they begin regaining health. In order to survive, agents must quickly forage for food, which is in limited supply, and water, which is infinitely renewable but only available at a smaller number of pools, in the presence of 100+ potentially hostile agents attempting to do the same. The starting and maximum quantities of food, water, and health, as well as associated loss and regeneration rates, are all configurable.

**Combat:** *This system is designed for direct competition among agents.* Agents can attack each other with three different styles – Range, Mage, and Melee. The attack style and combat stats of both parties determine accuracy and damage. This system enables a variety of strategies. Agents more skilled in combat can assert map control, locking down resource-rich regions for themselves. Agents more skilled in maneuvering can succeed through foraging and evasion. The goal is to balance between foraging safely and engaging in dangerous combat to pilfer other agents' resources and cull the competition. Accuracy, damage, and attack reach are configurable for each combat style.

**Skill Progression:** *This system is designed for long-term planning.* MMO players progress both by improving mechanically and by slowly working towards higher skill levels and better equipment. Policies must optimize not only for short-term survival, but also for strategic combinations of skills. In Neural MMO, foraging for food and water grants experience in the respective Hunting and Fishing skills, which enable agents to gather and carry more resources. A similar system is in place for combat. Agents gain levels in Constitution, Range, Mage, Melee, and defense through fighting. Higher offensive levels increase accuracy and damage while Constitution and Defense increase maximum health and evasion, respectively. Starting levels and experience rates are both configurable.

**NPCs & Equipment:** *This system is designed to introduce risk/reward tradeoffs independent from other learning agents.* Scripted non-playable characters (NPCs) with various abilities spawn throughout the map. Passive NPCs are weak and flee when attacked. Neutral NPCs are of medium strength and will fight back when attacked. Hostile NPCs have the highest levels and actively hunt nearby players and other NPCs. Agents gain combat experience and equipment by defeating NPCs, which spawn with armor determined by their level. Armor confers a large defensive bonus and is a significant advantage in fights against NPCs and other players. The level ranges, scripted AI distribution, equipment levels, and other various features are all configurable.

---

[4]It is a common misconception in RL that grid-worlds are fundamentally simplistic: Some of the most popular and complex MMOs on the market partition space using a grid and simply smooth over animations. These include RuneScape 3, OldScool Runescape, Dofus, and Wakfu, all of which have existed for 8+ years and maintained tens of thousands of daily players and millions of unique accounts

Table 1: Baselines on canonical SmallMaps and LargeMaps configs with all game systems enabled. Refer to Section 5 for definitions of Metrics and analysis to aid in Interpreting Results. The highest value for each metric is bolded for both configs.

| Model | Lifetime | Achievement | Player Kills | Equipment | Explore | Forage |
|---|---|---|---|---|---|---|
| **Small maps** | | | | | | |
| Neural Forage | 132.14 | 2.08 | 0.00 | 0.00 | 9.73 | 18.66 |
| Neural Combat | 51.86 | 3.35 | **1.00** | **0.27** | 5.09 | 13.58 |
| Scripted Forage | **252.38** | **7.56** | 0.00 | 0.00 | **37.07** | **26.18** |
| No Explore | 224.30 | 4.34 | 0.00 | 0.00 | 21.19 | 21.87 |
| Scripted Combat | 76.52 | 3.45 | 0.69 | 0.11 | 14.81 | 16.15 |
| No Explore | 52.11 | 2.72 | 0.73 | 0.18 | 9.75 | 14.35 |
| Scripted Meander | 28.62 | 0.08 | 0.00 | 0.00 | 4.46 | 11.49 |
| **Large maps** | | | | | | |
| Neural Forage | 356.13 | 2.75 | 0.00 | 0.00 | 10.20 | 18.02 |
| Neural Combat | 57.12 | 2.34 | **0.96** | 0.00 | 3.34 | 11.96 |
| Scripted Forage | **3224.31** | **28.97** | 0.00 | 0.00 | **136.88** | **47.83** |
| Scripted Combat | 426.44 | 10.75 | 0.71 | 0.04 | 53.15 | 24.60 |
| **Zero-shot** | | | | | | |
| S → L | 73.48 | 3.15 | 0.88 | 0.03 | 7.52 | 13.46 |
| L → S | 35.24 | 3.15 | 1.01 | 0.25 | 2.93 | 12.73 |

## 4 Models

**Pretrained Baseline:** We use RLlib's [22] PPO [23] implementation to train a single-layer LSTM [24] with 64 hidden units. Agents act independently but share a single set of weights; training aggregates experience from all agents. Input preprocessor and output postprocessor subnetworks are used to fit Neural MMO's observation and action spaces, much like in OpenAI Five and AlphaStar. Full architecture and hyperparameter details are available in the supplementary material. We performed only the minimal hyperparameter tuning required to optimize training for throughput and memory efficiency. Each experiment uses hardware that is reasonably available to academic researchers: a single RTX 3080 and 32 cores for 1-5 days – Up to 100k environments and 1B agent observations.

**Scripted Bots:** The Scripted Forage baseline shown in Table 1 implements a locally optimal min-max search for food and water over a fixed time horizon using Dijkstra's algorithm but does not account for other agents' actions. The Scripted Combat baseline possesses an additional heuristic that estimates the strength of nearby agents, attacks those it perceives as weaker, and flees from stronger aggressors. Both of these scripted baselines possess an optional Exploration routine that biases navigation towards the center of the map. Scripted Meander is a weak baseline that randomly explores safe terrain.

## 5 Baselines and Additional Experiments

Neural MMO provides small- and large-scale tasks and baseline evaluations using both scripted and pretrained models as canonical configurations to help standardize initial research on the platform.

**Learning Multimodal Skills:** The canonical SmallMaps config generates 128x128 maps and caps populations at 256 agents. Using a simple reward of -1 for dying and a training horizon of 1024 steps, we find that the recurrent model described in the last section learns a *progression of different skills* throughout the course of training (Fig. 4). Basic foraging and exploration are learned first. Average metrics for these skills drop later in training as agents learn to fight: the policies have actually improved, but the task has become more difficult in the presence of adversaries capable of combat. Finally, agents learn to selectively target passive NPCs as they do not fight back, grant combat experience, and drop equipment upgrades. This progression of skills occurs without any direct reward or incentive for anything but survival. We attribute this phenomenon to a *multiagent autocurriculum* [6; 25] – the pressure of competition incentivizes agents not only to explore (as seen in the first experiment), but also to leverage the full breadth of available game systems. We

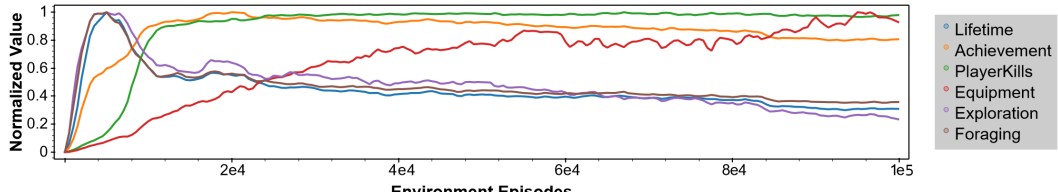

Figure 4: Agents trained on small maps only to survive learn a progression of skills: foraging and survival followed by combat with other agents followed by attacking NPCs to acquire equipment.

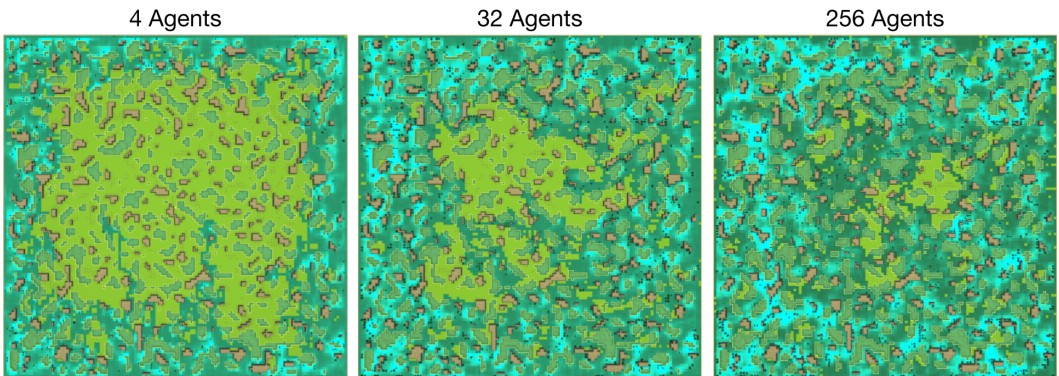

Figure 5: Competitive pressure incentivizes agents trained in large populations to learn to explore more of the map in an effort to seek out uncontested resources. Agents spawn at the edges of the map; higher intensity corresponds to more frequent visitation.

believe it is likely that continuing to add more game systems to Neural MMO will result in increased complexity of learned behaviors and enable even more new directions in multiagent research.

**Large-Scale Learning:** We repeat the experiment above using the canonical LargeMaps setting, which generates 1024x1024 maps and caps populations at 1024 agents. Training using the same reward and a longer horizon of 8192 steps produces a qualitatively similar result. Agents learn to explore, forage, and fight as before, but the policies produced are capable of exploring several hundred tiles into the environment – more than the SmallMaps setting allows. However, the learning dynamics are less stable, and continued training results in policy degradation.

**Zero-Shot Transfer:** We evaluated the LargeMaps policy zero-shot on the SmallMaps domain. It performs significantly better than random but worse than the agent explicitly trained for this domain. Surprisingly, transferring the SmallMaps model to LargeMaps domain performs better than the LargeMaps model itself but not as well as the scripted baselines.

**Metrics:** In Table 1, Lifetime denotes average survival time in game ticks. Achievement is a holistic measure discussed in the supplementary material. Player kills denotes the average number of other agents defeated. Equipment denotes the level of armor obtained by defeating scripted NPCs. The latter two metrics will always be zero for non-combat models. Explore denotes the average $L_1$ distance traveled from each agent's spawning location. Finally, Forage is the average skill level associated with resource collection.

**Interpreting Results:** The scripted combat bot relies upon the same resource gathering logic as the scripted foraging bot and has strictly more capabilities. However, since it is evaluated against copies of itself, which are more capable than the foraging bot, it actually performs worse across several metrics. This dependence upon the quality of other agents is a fundamental difficulty of evaluation in multiagent open-ended settings that we address in the Section 7. Our trained models perform somewhat worse than their scripted counterparts on small maps and significantly worse on large maps. The scripted baselines themselves are not weak, but they are also far from optimal – there is plenty of room for improvement on this task and much more on cooperative tasks upon which, as per our last experiment in Section 5, the same methods perform poorly.

**Domain Randomization is an Effective Curriculum:** Training on a pool of procedurally generated maps produces policies that generalize better than policies trained on a single map. Similar results have been observed on the Procgen environment suite and in OpenAI's work on solving Rubik's cube [5; 20]. The authors of the former found that different environments scale up to a different number of maps/levels. We therefore anticipate that it will be useful for Neural MMO users to understand precisely how domain randomization affects performance. We train using 1, 32, 256, and 16384 maps using the canonical 128x128 configuration of Neural MMO with all game systems enabled. Surprisingly, we only observe a significant generalization gap on the model trained with 1 map (Figs. 6 and Table 2). Note that we neglected to precisely control for training time, but performance is stable for both the 32 and 256 map models by 50k epochs. Interestingly, the 16384 map model exhibits different training dynamics and defeats NPCs to gather equipment three times more than the 32 map model. Lifetime increases early during training as agents learn basic foraging, but it then decreases as agents learn to fight each other.

It may strike the reader as odd that so few maps are required to attain good generalization where the same methods applied to Proc-Gen require thousands. This could be because every 128x128 map in Neural MMO provides spawning locations all around the edges of the map. Using an agent vision range of 7 tiles, there are around 32 spawning locations with non-overlapping receptive fields (and many more that are only partially overlapping). Still, this is only a total of 1024 unique initial conditions, and we did not evaluate whether similar performance is attainable with 8 or 16 maps. We speculatively attribute the remaining gap between our result and ProcGen's to the sample inefficiency of learning from rendered game frames without vision priors.

Figure 6: Lifetime curves over training on different numbers of procedurally generated maps. Each epoch corresponds to simulating one game map and all associated agents for 1024 timesteps.

**Population Size Magnifies Exploration:** We first consider a relatively simple configuration of Neural MMO in order to answer a basic question about many-agent learning: how do emergent behaviors differ when agents are trained in populations of various sizes? Enabling only the resource and progression systems with continuous spawning, we train for one day with population caps of 4, 32 and 256 agents. Agents trained in small populations survive only by foraging in the immediate spawn area and exhibit unstable learning dynamics. The competitive pressure induced by limited resources causes agents trained in larger populations to learn more robust foraging and exploration behaviors that cover the whole map (Fig. 5). In contrast, as shown by lower average lifetime in Table 4 in the supplementary material, increasing the test-time population cap for agents trained in small populations results in overcrowding and starvation.

**Emergent Complexity from Team Play:** Multi-population configurations of Neural MMO enable us to study emergent cooperation in many-team play. We enable the resource + combat systems and train populations of 128 concurrently spawned agents with shared rewards across teams of 4. Additionally, we had to disable combat among teammates and add an auxiliary reward for attacking other agents in order to learn anything significant. Under these parameters, agents learn to split into teams of 2 to fight other teams of 2 – not a particularly compelling or sophisticated strategy. Perhaps additional innovations are required in order to learn robust and general cooperation.

Table 2: Average lifetime during training on different numbers of maps and and subsequent evaluation on unseen maps.

| #Maps | Train | Test | Epochs |
|---|---|---|---|
| 1 | 66.08 | 52.73 | 67868 |
| 32 | 61.93 | 61.56 | 109868 |
| 256 | 62.67 | 61.91 | 58568 |
| 16384 | 52.30 | 53.25 | 99868 |

Table 3: Qualitative summary of related single-agent, multi-agent, and industry-scale environments. **Agents** is listed as the maximum for variable settings. **Horizon** is listed in minutes. **Efficiency** is a qualitative evaluation based on a discussion in the supplementary material.

| Environment | Genre | Agents | Horizon | Task(s) | Efficiency | Procedural |
|---|---|---|---|---|---|---|
| NMMO Large | MMO | 1024 | 85 | Open-Ended | High | Yes |
| NMMO Small | MMO | 256 | 10 | Open-Ended | High | Yes |
| ProcGen | Arcade | 1 | 1 | Fixed | Medium | Yes |
| MineRL | Sandbox | 1 | 15 | Flexible | Low | Yes |
| NetHack | Rougelike | 1 | Long! | Flexible | High | Yes |
| PommerMan | Arcade | 2v2 | 1 | Fixed | High | Yes |
| MAgent | - | 1M | 1 | Open-Ended | High | Partial |
| DoTA 2 | MOBA | 5v5 | 45 | Fixed | Low | No |
| StarCraft 2 | RTS | 1v1 | 20 | Fixed | Low | No |
| Hide & Seek | - | 3v3 | 1 | Flexible | Low | Yes |
| CTF | FPS | 3v3 | 5 | Fixed | Low | Yes |

# 6 Related Platforms and Environments

Table 3 compares environments most relevant to our own work, omiting platforms without canonical tasks. Two nearest neighbors stand out. MAgent [26] has larger agent populations than Neural MMO, NetHack [10] has longer time horizons, and both are more computationally efficient. However, MAgent was designed for simple, short-horizon tasks and NetHack is limited to a single agent. Several other environments feature a few agents and either long time horizons or high computational efficiency, but we are unaware of any featuring large agent populations or open-ended task design

**OpenAI Gym:** A classic and widely adopted API for environments [27]. It has since been extended for various purposes, including multiagent learning. The original OpenAI Gym release also includes a large suite of simple single-agent environments, including algorithmic and control tasks as well as games from the Atari Learning Environment.

**OpenAI Universe:** A large collection of Atari games, Flash games, and browser tasks. Unfortunately, many of these tasks were much too complex for algorithms of the time to make any reasonable progress, and the project is no longer supported [4].

**Gym Retro:** A collection of over 1000 classic console games of varying difficulties. Despite its wider breadth of tasks, this project has been largely superseded by the ProcGen suite, likely because users are required to provide their own game images due to licensing restrictions [3].

**ProcGen Benchmark:** A collection of 16 single-agent environments with procedurally generated levels that are well suited to studying generalization to new levels [5].

**Unity MLAgents:** A platform and API for creating single- and multi-agent environments simulated in Unity. It includes a suite of 16 relatively basic environments by default but is sufficiently expressive to define environments as complex as high-budget commercial games. The main problem is the inefficiency associated with simulating game physics during training, which seems to be the intended usage of the platform. This makes MLAgents better suited to control research [11].

**Griddly:** A platform for building grid-based single- and multi-agent game environments with a fast backend simulator. Like OpenAI Gym, Griddly also includes a suite of environments alongside the core API. It is sufficiently expressive to define a variety of different tasks, including real-time strategy games, but its configuration system does not include a full programming language and is ill-suited as backend for in Neural MMO [14].

**MAgent:** A platform capable of supporting one million concurrent agents per environment. However, each agent is a simple particle operating in a simple cellular automata-like world. It is well suited to studying collective and aggregate behavior, but has low per-agent complexity [26].

**Minecraft:** MALMO [28] and the associated MineRL [15] Gym wrapper and human dataset enable reinforcement learning research on Minecraft. Its reliance upon intuitive knowledge of the real world makes it more suitable to learning from demonstrations than from self-play.

**DoTA 2:** Defense of the Ancients, a top esport with 5v5 round-based games that typically last 20 minutes to an hour. DoTA is arguably the most complex game solved by RL to date. However, the environment is not open source, and solving it required massive compute unavailable to all but the largest industry labs [17].

**StarCraft 2** A 1v1 real time strategy esport with round-based games that typically last 15 minutes to an hour. This environment is another of the most complicated to have been solved with reinforcement learning [18] and is actually open source. While the full task is inaccessible outside of large-scale industry research, the StarCraft Multi-Agent Challenge provides a setting for completing various tasks in which, unlike the base game of StarCraft, agents are controlled independently [29].

**Hide and Seek:** Laser tag mixed with hide and seek played with 2-3 person teams and ~1 minute rounds on procedurally generated maps [6].

**Capture the Flag:** Laser tag mixed with capture the flag played with 2-3 person teams and ~5 minute rounds on procedurally generated maps [30].

**Pommerman** A 4 agent environment playable as free for all or with 2 agent teams. The game mechanics are simple, but this environment is notable for it's centralized tournament evaluation that enables researchers to submit agent policies for placement on a public leaderboard. It is no longer officially supported [13].

**NetHack** By far the most complex single-agent game environment to date, NetHack is a classic text-based dungeon crawler featuring extended reasoning over extremely long time horizons [10].

**Others:** Many environments that are not video games have also made strong contributions to the field, including board games like Chess and Go as well as robotics tasks such as the OpenAI Rubik's Cube manipulation task [20] which inspired NeuralMMO's original procedural generation support.

## 7    Limitations and Discussion

**Many problems do not require massive scale:** Neural MMO supports up to 1024 concurrent agents on large, 1024x1024 maps, but most of our experiments consider only up to 128 agents on 128x128 maps with 1024-step horizons. We do not intend for every research project to use the full-scale version – there are many ideas in multi-agent intelligence research, including those above, that can be tested efficiently at smaller scale (though perhaps not as effectively at the very small scale offered by few-agent environments outside of the Neural MMO platform). The large-scale version of Neural MMO is intended for testing multiple such ideas in conjunction and at scale – as a sort of "realistic" combination of multiple modalities of intelligence. For example, team-based play is interesting in the smaller setting, but what might be possible if agents have time to strategize over multiple hours of play? At the least, they could intentionally train and level their combat skills together, seek out progressively more difficult NPCs to acquire powerful armor, and whittle down other teams by aggressively focusing on agents at the edge of the pack. Developing models and methods capable of doing so is an open problem – we are unaware of any other platform suitable for exploring extended socially-aware reasoning at such scale.

**Absence of Absolute Evaluation Metrics:** Performance in open-ended multiagent settings is tightly coupled to the actions of other agents. As a result, average reward can decrease even as agents learn better policies. Consider training a population of agents to survive. Learning how to forage results in a large increase to average reward. However, despite making agents strictly more capable, learning additional combat skills can decrease average lifetime by effectively making the task harder – agents now have to contend with hostile potential adversaries. This effect makes interpreting policy quality difficult. We have attempted to mitigate this issue in our experiments by comparing policies according to several metrics instead of a single reward. This can reveal learning progress even when total reward decreases – for example, in Figure 4, the Equipment stat continues to increase throughout training. More recently, we have begun to shift our focus towards tournament evaluations that abandons absolute policy quality entirely in favor of a skill rating relative to other agents. This approach has been effective thus far in the competition, but we anticipate that it may not hold for all environment settings as Neural MMO users continue to innovate on the platform. With this in mind, we believe that developing better evaluation tools for open-ended settings will become an important problem as modern reinforcement learning methods continue to solve increasingly complex tasks.

# 8 Acknowledgements

This project had been made possible by the combined effort of many contributors over the last four years. Joseph Suarez is the primary architect and project lead. Igor Mordatch managed and advised the project at OpenAI. Phillip Isola advised the project at OpenAI and resumed this role once the project residence shifted to MIT. Yilun Du assisted with experiments and analysis on v1.0. Clare Zhu wrote about a third of the legacy THREE.js web client that enabled the v1.0 release. Finally, several open source contributors have provided useful feedback and discussion on the community Discord as well as direct bug fixes and features. Additional details are available on the project website.

This work was supported in part by: Andrew (1956) and Erna Viterbi Fellowship Alfred P. Sloan Scholarship (G-2018-10127)

Research was also partially sponsored by the United States Air Force Research Laboratory and the United States Air Force Artificial Intelligence Accelerator and was accomplished under Cooperative Agreement Number FA8750-19-2-1000. The views and conclusions contained in this document are those of the authors and should not be interpreted as representing the official policies, either expressed or implied, of the United States Air Force or the U.S. Government. The U.S. Government is authorized to reproduce and distribute reprints for Government purposes notwithstanding any copyright notation herein.

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
