# OpenReview forum: "The Neural MMO Platform for Massively Multiagent Research"
_NeurIPS.cc/2021/Track/Datasets_and_Benchmarks/Round1 — NeurIPS 2021 Datasets and Benchmarks Track (Round 1)_

### Official Review · Reviewer_jSbC · 2021-06-24
**Actively developed large-scale multi-agent environment; great potential but further polishing still needed**

**Rating:** 8
**Confidence:** 4
**Correctness:** All claims appear sound.

**Strengths:**

I believe this project has great potential to be of value for multi-agent learning research, in particular research into population training and emergence of social behaviour. I also gladly observe that Neural MMO is very actively developed and has a detailed documentation which clearly much work has been put into.

The inclusion of a discussion on remaining limitations of Neural MMO is also very useful. However, it would also be useful to concretely state for which research the authors envision Neural MMO to be a valuable environment for.

The description of the core environment features throughout Section 2 and 3 are clear and additional information is made available through the online documentation.

The set of initial experiments highlight some challenges for training agents in Neural MMO which are further highlighted by the authors and some visualisation tools are demonstrated.

**Weaknesses:**

Overall, I believe this project has great potential but its usefulness largely depends on the simplicity of adopting this environment for research. Despite the extensive documentation, there still appears a significant barrier of entry. The documentation helps, but due to the complexity it is still difficult to understand how to use Neural MMO. I strongly suggest to add exemplary code in the documentation to showcase how certain use-cases could be achieved, such that the user can gradually understand the framework and environment from there without needing to read pages upon pages of documentation. Code snippets as shown in Figure 2 are very helpful (but need to be verified - see below for more details) !

Below, I provide some more detailed comments on several aspects of this work.



**Writing**

This work frequently refers to the game genre of MMOs. While such references and comparisons can help to motivate certain aspects, they often appear irrelevant (e.g. refering to design philosophy of classic MMOs in ll. 108ff, tile-based MMOs in ll. 126ff and skill progression in ll. 144f) . Simply explaining these concepts in the context of Neural MMO might help the reader to follow, in particular if they had no prior exposure to MMO games.

In the discussion on metrics (Section 5), it is stated that "Achievement" is discussed in the supplementary material (l. 201f). It would be helpful if the paper could at least provide a brief intuition into what this metric is meant to represent before referring to the supplementary material for more details.



**Experiments**

For trained baselines, the framework and algorithm is explained in Section 4. However, the training states a single-agent RL training setup whereas the tasks contain 1. multiple agents of 2. varying quantity, so it appears unclear how all agents are controlled and how the policy is optimised. Is the trained policy used to control all agents, so parameters are effectively shared across all agents which follow the same policy? Is the experience of all agents used to train this model using PPO? Or is each agent trained independently using the described method? If latter, how are newly spawned agents initialised?

As discussed in Section 7, a suitable evaluation metric is still missing and currently being developed. Such a metric is crucial for adoption as non-expert users of the environment might not be able to otherwise differentiate between decreasing lifetime due to well-trained competition or failure in learning such as catastrophic forgetting.



**Environment design and interface**

I agree that a benchmark environment which combines a variety of challenges of multi-agent learning has value as a challenge task to identify whether trained agents are able to combine such skills successfully. However, the majority of research still focuses on individual challenges due to the complexity of multi-agent learning. It is stated that Neural MMO features "modular game systems that are individually toggleable and configurable" (l. 28f) but this appears to be not further discussed in the paper or supplementary material (I was unable to find more information in the NeurIPS-specific material but have not read the entire (very detailed) documentation of this project). Having the ability to decide which challenges to include would significantly improve applicability of this environment for research.

Secondly, ease-of-use for such a project appears essential to be adopted for research. I appreciate the development of a gym-like interface which helps but I was unable to run the sample code provided in Figure 2 (left). I followed all installation instructions stated in the documentation on a Ubuntu 20.04 system but was unable to import the environment without a error (might relate to existing issue in Github repository).

**Additional Feedback:**

In the supplementary material webpage, there are few miss-typed commands and typos

- In Author Statement: "textbf{neuralmmo.github.io}"
- In Source Material: " They simpler than the real world" --> They **are** simpler ...
- In Achievement System: "(Fig. ref{achievements})"
- In Serialization: "This allows us to maintain a table where each row is an object and each attribute is an attribute." --> ... and each **column** is an attribute. (?)
- In Designing for Efficient Complexity: "textit{efficiency} and textit{complexity}."

UPDATED SCORE FOLLOWING AUTHORS REBUTTAL AND IMPROVEMENTS

**Clarity:**

Overall, the paper is well written with some clarity concerns regarding figures:

- Figure 1: Text in interaction visualization stack is hard to read. Suggest to increase the contrast of text and layers to improve readability.
- Figure 3:
    - labels of ticks on x-axis use 3 decimal digits which are 0 throughout (just use 0e4, 2e4, ...).
    - label of last x-tick is cut off (1.000e+) --> 1e+5 (presumably?)
- Figure 4: What are colours refering to? It is not clear which colour(s) correspond to exploration in which way.
- Figure 6: This should be a table



Also, the NeurIPS style guide states that "The table number and title always appear before the table" --> place title captions above tables.

Table 1:
- Caption should state what bold values indicate.
- Clearly separate and/or highlight trained agents and scripted baselines.


**Documentation:**

Excellent documentation is provided through the supplementary material / project webpage.


**Ethics:**

I foresee no ethical concerns regarding this project which require further discussion.


**Relation To Prior Work:**

Similarities and differences to existing environments are largely stated sufficiently.

However, comparison to multi-agent environments appears underrepresented. Suitable candidates which should be considered for comparisons are

- Starcraft Multi-Agent Challenge [1]
- MALMO [2]
- Deepmind Lab [3]
- Multi-Agent Particle Environment [4, 5]



[1] M. Samvelyan, T. Rashid, C. Schroeder de Witt, G. Farquhar, N. Nardelli, T.G.J. Rudner, C.-M. Hung, P.H.S. Torr, J. Foerster, S. Whiteson. The StarCraft Multi-Agent Challenge, CoRR abs/1902.04043, 2019.

[2] Johnson M., Hofmann K., Hutton T., Bignell D. (2016) The Malmo Platform for Artificial Intelligence Experimentation. Proc. 25th International Joint Conference on Artificial Intelligence, Ed. Kambhampati S., p. 4246. AAAI Press, Palo Alto, California USA. https://github.com/Microsoft/malmo

[3] Beattie, Charles, Joel Z. Leibo, Denis Teplyashin, Tom Ward, Marcus Wainwright, Heinrich Küttler, Andrew Lefrancq et al. "Deepmind lab." *arXiv preprint arXiv:1612.03801* (2016).

[4] Mordatch, Igor, and Pieter Abbeel. "Emergence of Grounded Compositional Language in Multi-Agent Populations. CoRR abs/1703.04908 (2017)." arXiv preprint arXiv:1703.04908 (2017).

[5] Lowe, Ryan, Yi Wu, Aviv Tamar, Jean Harb, Pieter Abbeel, and Igor Mordatch. "Multi-agent actor-critic for mixed cooperative-competitive environments." arXiv preprint arXiv:1706.02275 (2017).

**Summary And Contributions:**

This paper presents a new environment for multi-agent research focused on long time horizons, large groups of agents and hard modular tasks which require agents to combine a variety of skills to be successful. Alongside the open-source environment, a set of tools for visualisations and logging are released and extensive documentation has been made available. The environment appears to be actively developed and the authors promise continual development and support. Initial experiments demonstrate the effectiveness of large populations for training agents in the Neural MMO environment and discuss challenges for multi-agent learning.

---

### Official Review · Reviewer_HkHe · 2021-07-04
**A new multiagent platform for RL with a pledge of continued support**

**Rating:** 7
**Confidence:** 3
**Correctness:** Claims appear correct.
**Clarity:** Yes.

**Strengths:**

(1) The Neural MMO motivation makes sense and is compelling.
(2) Game systems are well described and the examples of config files are easy to follow.
(3) Experiments are interesting. The utilization of small map experiments will be good to democratize research for multi agent RL problems across.
(4) The addition of procedural generation is good and should lead to much more robust policies. The existence of this generation should help researchers create better algorithms for this setting.
(5) A believable pledge of support. I hope this work will garner active community participation and interest over the next few years.

**Weaknesses:**

(1) Visualizations are hard to understand and do not convey much information (figure 1/2).
(2) Related work section needs a final summary paragraph that conveys where Neural MMO is situated in the field.
(3) MMO is never expanded to its full form as far as I can tell. The introduction should have a sentence or two describing what a conventional MMO is.

**Additional Feedback:**

None.

**Documentation:**

Appears satisfactory and authors appear committed too maintaining and improving their work.

**Relation To Prior Work:**

A lot of prior work is mentioned. The related work section needs a summar paragraph that explicitly places Neural MMO in the context of all these related works.

**Summary And Contributions:**

The authors introduce Neural MMO, a new research platform for multi-agent RL and learning. Neural MMO is open sourced and easily available providing many benefits that are currently hard to find in the field of RL including large agent populations (upto 1024!), very long time horizons etc. Editing neural MMO appears simple through the use of config files and support exists for popular baselines. The varying games systems that drive the MMO are described in detail and as a whole the paper should as a good starting point for people who are interesting in using this work for their own research.

---

### Official Review · Reviewer_FoXt · 2021-07-06
**A nice platform for multi-agent research but not quite novel**

**Rating:** 6
**Confidence:** 3
**Correctness:** I agree with the claimed made in this…

**Strengths:**

The user API and documentation is very friendly to users. I think this platform is also efficient enough for large scale multi-agent learning. In general, this is a good platform to study multi-agent learning.

**Weaknesses:**

1. Different types of agent can be added. Currently all agents have the same action space. Having different action space can potentially leads to interesting team play.
2. In this game, all the agent have the same goal, which could possibly lead to very similar behaviors among agents.
3. I agree with the author about the claim of not all the problem require large scales.

**Additional Feedback:**

For the PPO baseline, are all the agents using central policy? It would be interesting to see diverse agent behaviors instead of all agent having the same behavior in such an environment.


**Clarity:**

The paper is well-written and the document of this platform actually looks very well-explained too.

**Documentation:**

The codes and documents are public and very accessible.

**Ethics:**

I don't have any ethical concern.

**Relation To Prior Work:**

This paper clearly show how Neural MMO is different from other game platform used for multi-agent research.

**Summary And Contributions:**

This work present a large scale environment for multi-agent research called Neural MMO. This environment provide functions for map generation and environment customization, logging for others' ease of use. The platform is efficient to collect multi-agent data and accessible for multi-agent research. The purposed environments (or game system) includes various resources for agents and the agents have a competition relationship with each others. The experimental results show that the agent can learn different skills s throughout the course of training, which is quite interesting. The authors also show the emergent of team play using their multi-population configuration.

---

### Decision · Program_Chairs · 2021-07-26

**Decision:**

Accept

**Comment:**

All reviewers recommend acceptance, and I follow this unanimous recommendation.